# Screening for Psychosocial Distress: A Brief Review with Implications for Oncology Nursing

**DOI:** 10.3390/healthcare12212167

**Published:** 2024-10-31

**Authors:** Margaret I. Fitch, Irene Nicoll, Stephanie Burlein-Hall

**Affiliations:** 1Bloomberg Faculty of Nursing, University of Toronto, Toronto, ON M5T1P8, Canada; 2Healthcare Consultant, Toronto, ON, Canada; nicollirene4@gmail.com; 3Education Consultant, Ajax, ON, Canada; stephanie@burlein.com

**Keywords:** cancer care, screening for distress, oncology nursing, psychosocial oncology

## Abstract

Purpose: Psychosocial care is an integral component of caring for individuals living with cancer. The identification of psychosocial distress has been acknowledged as a hallmark of quality cancer care, and screening for distress standards has been established in several countries. The purpose of this brief review is to highlight recent developments in screening for distress in oncology populations; to provide insight into significant trends in research and implementation; and to explore implications for oncology nursing practice. Methods: This paper reports a brief review of the literature from March 2021 to July 2024 on the topic of screening for distress in oncology. The literature was accessed through PubMed and reviewed by two authors. Trends in the topics presented were identified independently and then discussed to achieve consensus. Results: The search within the designated period produced 47 publications by authors in North America, Australasia, and Europe. Topic trends included the design and adaptation of tools for special populations, the use of technology, descriptions of programs, identification of benefits, challenges, and overcoming barriers to screening for distress. Conclusions: Screening for distress is endorsed as part of the provision of quality oncology care. Nurses have an important role in screening individuals at risk for developing psychosocial problems and acting to reduce the associated morbidity. By continuing to be informed and educated about the emerging developments in screening for distress, nurses can understand and overcome barriers to implementation.

## 1. Introduction

Cancer and its treatment can have a profound impact on individuals diagnosed with the disease and their family members [1,2]. Experiencing cancer creates physical, emotional, and practical challenges that can generate psychosocial distress for individuals [3]. As many as 88% of cancer patients have reported heightened levels of distress [4], including anxiety, depression, changes in body image, fear of recurrence, loneliness, and existential angst. Should this distress remain unidentified and untreated, it can decrease quality of life [5,6] and ultimately impact morbidity and mortality [7,8].

### 1.1. Background

The early identification of psychosocial distress has been acknowledged as a hallmark of quality cancer care [9]. Standards were established in Canada in 2008 [10] and the United States in 2012 [11], outlining requirements for cancer programs to have a system or procedure to screen cancer patients for psychosocial distress. Over the past two decades, screening for distress has been supported by various cancer leaders and organizations around the world [12] as a first step in the process of identifying and managing symptoms and emotional distress. Oncology nurses have been identified as the ideal healthcare providers to perform screening for distress. [13,14]. Early writings related to screening for distress described issues that patients experienced, which resulted in psychosocial distress [15,16,17], and the implications for individuals if this distress was not identified and managed [18,19]. Distress, if untreated, can lead to increased hospitalization, visits to emergency departments, and physicians’ offices, as well as the escalation of turmoil experienced by cancer patients and impact on quality of life. Individuals with high distress have trouble making treatment decisions and following treatment regimes.

Advocates called for the early, pre-emptive identification of those at risk [20] so that preventative measures could be taken. As understanding about implementing screening for distress deepened, authors began emphasizing the importance of using standardized measurement screening tools for consistency and comparability [21]. Tools such as the Distress Thermometer [22], the Edmonton Symptom Assessment System [23], the Canadian Problem Checklist [21], and the Hospital Anxiety and Depression Scale [24] were recommended as tools that could be easily applied and interpreted in busy clinical settings [25,26].

The early initiatives to implement screening for distress quickly revealed that the administration of a screening tool alone was insufficient to affect outcomes for patients. Screening needs to be followed by timely interpretation of scores and a follow-up action based on the scores [27,28], including a conversation with the patient about what course of action would be desirable [29]. Referral algorithms and clinical pathways were introduced to assist staff in responding both rapidly and appropriately [30]. Without timely follow-up, patients soon declined participation in screening because they did not see their perspectives resulting in a response.

Ideally, a cascade or series of actions characterizes a programmatic approach to screening for distress [31]. Implementing screening for distress often requires changes in healthcare professionals’ practice and workflow. Implementation is a complex process [32], and cancer centers have found that there are barriers and challenges to the successful implementation of screening for distress [33,34]. Over the years, various published reports have described critical success factors and strategies for successful approaches to its implementation [27,35,36,37,38,39].

Standards for oncology nursing practice describe expectations for providing psychosocial care to cancer patients and their family members [40]. Incorporating screening for distress expectations within a daily oncology nursing practice can be a beneficial strategy [41]. The approach can assist in the early identification of symptoms and emotional distress and set the stage for interventions that would influence the mental health of patients and family members. However, nurses have identified discomfort and a lack of confidence in assessing psychosocial distress and recommending interventions [42,43]. Often, targeted education is required to enhance their psychosocial assessment and intervention knowledge and skills [44].

### 1.2. Purpose

The purpose of this brief review was to highlight recent significant developments in screening for psychosocial distress in cancer populations. The aim was to gain insight into current trends regarding research and the implementation of screening for distress to understand the implications for oncology nursing practice and the mental healthcare of individuals with cancer.

## 2. Methods

This paper reports a brief review of the literature from the past three and a half years on the topic of screening for distress in oncology. The review search was guided by using the following keywords and phrases together with ‘cancer’: ‘screening for distress’, screening for psychosocial distress’, ‘implementing screening for distress’, and ‘barriers and facilitators to screening for distress’. A search of Medline via PubMed using these words identified relevant publications in English during the period between March 2021 and July 2024. All article design types were considered (e.g., reviews, perspectives papers, descriptive/intervention studies), provided that the content focused on the activity or procedure of screening for psychosocial distress using a standardized/validated instrument. Articles with a focus only on reporting levels of psychosocial distress in a patient sample, symptoms of distress, or interventions to manage distress were not included.

All articles meeting the inclusion criteria were included and read by two authors (MF, IN). Each selected article was reviewed to identify the country of origin, the population targeted, the screening instrument utilized, and significant findings reported. A quality analysis was not performed. The extracted results were grouped into broad topic areas and conceptualized as current trends in screening for distress being written about in the literature. All authors discussed and reached consensus regarding the overall trends. Each of the trends is summarized below, presenting significant developments regarding research and the implementation of screening for distress in oncology settings. Implications of these results for oncology nursing practice are subsequently discussed.

## 3. Results

The search within the designated period produced a total of 47 publications. These publications were written by authors based in North America (n = 28), Australasia (n = 13), and Europe (n = 6) (see Table 1). The majority of the articles described research projects (n = 42), while two described screening for distress programs, and three were commentaries. Of the 33 articles that described using a tool to measure distress, the Distress Thermometer and its accompanying Problem Checklist were mentioned most frequently (n = 24). Other tools were either newly developed electronic applications for distress screening (n = 2), the use of validated tools such as the Edmonton Symptom Assessment System (ESAS), the Hospital Anxiety and Depression Scale (n = 6), or a new tool for cancer survivor use (n = 1). Articles were published in a variety of journals, with the most appearing in the Journal of Psycho-Oncology (n = 7), Supportive Care in Cancer (n = 5), the Journal of Clinical Oncology (Clinical Practice) (n = 4), the Journal of Psychosocial Oncology (n = 4), and Cancer (MDPI) (n = 2).

### 3.1. Trends Reported in Publications Concerning Screening for Distress

Analysis of the topics in the selected articles revealed several significant trends related to screening for psychosocial distress. These trends included designing and adapting tools (psychometric testing, special populations, technology), describing programs (models), the identification of challenges (barriers and facilitators), identifying screening benefits, and recommendations for improving screening for distress. As a practice intervention, there has now been sufficient effort to implement screening for distress approaches so that there are clear opportunities to evaluate and better understand the benefits or drawbacks of screening for distress. Each trend will be highlighted below.

#### 3.1.1. Trend: Designing and Adapting Tools

##### Testing Tools—Psychometric Evaluation

Psychometric evaluation of screening tools continues to be reported. It is important to assess the performance of a new tool or a tool that has been adapted because reliability and validity can be influenced by context of the setting or characteristics of the population. For example, Al-Shaaobi et al. [45] assessed the performance of the Distress Thermometer when used with in-patients, and Reid et al. [46] reported on observed differences in referral support services with Hispanic and Black cancer patients. A particular psychometric concern across several papers was validating the cut-off point for identifying high distress and thus, knowing when referrals should be offered. Sun et al. [47] performed a meta-analysis and determined an optimal (pooled sensitivity with 2851 patients) cut-off of four on the Distress Thermometer for Asian Patients.

##### Special Populations

An emerging trend is the testing and validation of the Distress Screening Thermometer, its accompanying Problem Checklist generally, and its modification for specific patient populations, cancer survivors, and caregivers.

Several articles focused on head and neck cancer patients [48,49,50,51], considered the most emotionally distressing of all cancers due to the impact on basic functioning and physical appearance [52]. The Problem Checklist for this group was adapted to include issues such as speech, swallowing, and disfigurement [48]. Other adaptations for specific populations included gynecological cancer patients (adding leg edema, changes in urination, and fertility issues) [53]; adolescent and young adult cancer patients (isolation from friends, guilt, boredom, fitness/sporting ability) [54]; and, for those at risk of financial hardship, spending on medical bills, lost savings, and accumulation of debt were included [48]. Many of the studies engaged patients and survivors in the modification of screening tools to highlight the issues they identified as significantly important.

The pediatric population is another growing, vulnerable group receiving attention [55]. Children undergoing treatment for cancer may have multiple medical procedures and clinic visits as well as experiences of side effects and unrelieved symptoms [56]. Screening tools designed specifically for pediatric cancer patients were tested and evaluated for the efficacy and effectiveness in revealing and resolving symptoms of distress [55]; the electronic administration of screening and timely integration of results into the electronic health record in real time were shown to be critical factors in the successful adoption of e-screening [57]; and the refinement of a two-step screening process was shown to reduce false positives [58].

The use and limitations of screening for distress for economic and/or ethnic and racial disparity among gynecological, endometrial, lung and ovarian cancer patients, and survivors regarding access to various support services were also examined. Liang et al. [59] found that over a third of gynecological patients with financial hardship were not identified with routine screening without the addition of dedicated screening tools for financial hardship. Maldonado et al. [60] observed that those with positive distress screens were more likely to report financial hardship but that over half of those who experienced financial hardship related to cancer were missed. Rohan et al. [61] examined the disparities in screening rates among lung and ovarian cancer survivors of different ethnic and cultural groups, and Reid et al. [46] hypothesized that screening tools may not adequately measure distress in non-white populations given the “historical forces and discrimination that have shaped interactions” with the medical system.

Studies from Asia focused on the validation of the Distress Thermometer and Problem Checklist. Three studies concluded that the Distress Thermometer with a cut off score of four, with its accompanying Problem Checklist, were effective tools in assessing and addressing both inpatient and outpatient needs [45,47,62]. Hirayama et al. [63] created and validated a tool specifically tailored for the Japanese AYA population that was then trialed as an electronic screening tool [64]. Results were automatically included in patients’ electronic medical records that led to quicker responses by the multi-disciplinary team.

Two articles highlighted screening for distress in cancer caregivers. These individuals have profound emotional bonds with patients and often perform key roles in ensuring family solidarity, providing emotional and practical assistance as well as managing their personal levels of anxiety and depression [65,66]. Manikowski et al. [67] found that the use of an electronic screening tool resulted in identifying close to 17% of caregivers who had elevated levels of distress and who required referrals. Zaleta et al. [68] studied and refined a web-based caregiver screening tool designed to identify the unmet needs of cancer caregivers and connect those in need of resources and support. Elements included caregiver wellbeing, patient wellbeing, caregiving tasks, finances, and health/lifestyle. The tool was considered a reliable and valid measure of caregiver distress.

The Distress Thermometer and its accompanying Problem Checklist were validated in these studies, examined for relevance to, or modified for special populations. The importance of regular and rapid distress assessments throughout the cancer journey was highlighted to identify and address high levels of distress. These articles emphasize the need for screening in special populations to proactively identify and address the needs of vulnerable cancer patients and overcome barriers. Further research regarding the reasons that patients, survivors, and caregivers often declined referrals and the potential impact of lack of accessibility and availability of services is required.

##### Technology

Several articles highlighted the increasing creation, use, and validation of electronic screening tools for use outside clinical settings, tailored for cancer patients, survivors or caregivers, and/or by healthcare professionals working in telemedicine. The work described reflects the growing types of electronic screening for distress tools and the effectiveness, challenges, and limitations that the use of virtual tools can present.

Several articles described the use of screening tools in telemedicine. DeGuzman et al. [48] reviewed the introduction of a telemedicine-delivered screening intervention to assess cancer-related distress in rural head and neck patients. While the intervention identified those with high distress and recommended support services, the referral uptake was found to be low. LaCouteur et al. [69] examined transcripts of calls by nurses responding to a cancer helpline. The authors found that, when screening questions were only mentioned casually near the end of the call, callers were less likely to choose to participate. When raised as the caller’s distress levels emerged naturally in the conversation, callers were more likely to respond positively, resulting in a greater number of referrals. Finally, Seib et al. [70] described a study comparing the effectiveness of a single-session nurse self-management and a five-session psychologist cognitive behavioral intervention for patients calling into a cancer helpline. Those that offered repeat screenings demonstrated progressively lower distress scores.

Bultz and Watson [71] found that the completion of screening for distress assessments dropped from 70% to 15% for patients participating in virtual rather than in person visits due to restrictions related to COVID-19. Both referrals and patient satisfaction with emotional support also dropped for virtual patients. Sutton et al. [72] studied participation in pre-appointment electronic screening and found that older patients, non-white and of lower socioeconomic status, were less likely to have access to and/or to be willing to engage in electronic screening yet may be more likely to experience distress.

Three articles describe the development, refinement, and/or evaluation of electronic screening tools for pediatric patients, pediatric caregivers, and cancer caregivers. Marchak et al. [57] described the efforts of pediatric oncology professionals to adapt and promote the Distress Thermometer for pediatric oncology outpatients. Manikowski et al. [67] evaluated an electronic screening tool for pediatric patients and their caregivers. Analysis showed that a psychosocial screening program for pediatric caregivers linked to a facility’s electronic health record system was effective in responding to caregiver distress and can be adapted to reach families across racial, ethnic, and language groups. Another study conducted by Zaleta et al. [68] described the refinement of the screening tool component of a web-based support program designed for cancer caregivers. Its psychometric properties demonstrated reliability, validity, and consistency as measures of distress.

These articles refer to the successes and limitations of virtual screening for distress tools in telemedicine delivered by healthcare providers. When introduced effectively, screening for distress can be an important tool to identify those in need of assistance as well as highlight the importance of awareness of the availability and accessibility of services. Given that efficient, electronic screening may disadvantage some groups, healthcare facilities need to ensure that all patients have access to screening.

#### 3.1.2. Trend: Describing Programs

Several articles described models for programmatic approaches to screening for distress. The programmatic approaches included a series of steps for screening with the use of standardized tools and actions to take based on scores generated from the tools. All focused on early and on-going identification of distress and the need to link the identification to referral and follow-up by psychosocial experts [47,73,74,75]. Incorporating patient screening and the response to the scores routinely in daily practice as part of the regular workflow was recommended [51,76,77]. Screening tools need to be relevant to the populations [53,54,60,69]. Where available, the initial steps ought to be incorporated into the electronic patient record so that clear documentation is evident. An interdisciplinary team approach and collaboration among departments are seen as making the referral and uptake easier for patients [74,78,79,80]. A tiered or stepped approach to psychosocial referral and intervention is recommended as a more tailored response for patients [70,81,82] as well as a cost-efficient model [83]. Authors also identified the need for clearly stated role expectations and responsibilities for screening and follow-up, which are promoted by the leadership of the health facility [76,84]. However, only one article clearly identified nurses as responsible for initial screening [85], and little description of specific time points for screening was included. Finally, linking clearly defined clinical pathways of referrals to screening results and collaboration with interdisciplinary team members promotes appropriate follow-up intervention [85,86]. Because this brief review focused on the procedures of screening for psychosocial distress, articles describing follow-up interventions to manage distress were not included. The importance of follow-up and relevant interventions, however, was emphasized throughout the various articles describing programmatic approaches.

#### 3.1.3. Trend: Identifying Challenges with Screening and Recommendations for Improvement

The expanding experience with implementing screening for distress has fostered identification and insight into the challenges of implementing a program. Many articles described barriers and facilitators to implementation identified either through administrative datasets [72,87,88] or qualitative interviews [76,77,89]. In the qualitative studies, staff members who had been using screening in their daily practices reiterated the need for the psychosocial care of patients and the potential benefit of screening for distress. However, numerous barriers were described that could interfere with the effectiveness of screening approaches, including those related to patients, clinicians, teams, institutions, screening tools and guidelines, and policy. Most frequently identified barriers were the lack of awareness and the lack of education about psychosocial distress and screening, the lack of resources (personnel and equipment), and the lack of clarity surrounding screening protocols and role expectations. The specific barriers reported in the articles are listed in Table 2. Facilitators were often identified as strategies which can overcome the barriers. For example, the lack of senior leadership was a barrier, while having a supportive senior leader was seen as a facilitator [76]. A helpful new tool was designed by Simnacher et al. [90] to assess barriers to screening prior to implementation. It is a brief 14-item questionnaire that is meant to be completed by staff members who will be responsible for implementing a program of screening for distress.

Authors frequently described similar strategies for improving screening for distress approaches. Many of the recent articles focus on actions required after the patient completes the screening tool for the program to be successful [86,89]. Emphasis was placed on the importance of screening initiatives accompanied by practice change and tailoring the program approaches for the specific facility locations, workflows, and patient populations. For example, Pang et al. [82] identified the need for a specific screening tool for survivors calling a community-based volunteer agency while Rohan et al. [61], Reid et al. [46], and Sutton et al. [72] described the need for using approaches that were relevant for race and ethnicities. Table 3 lists the commonly identified recommendations for improvement in screening based on the recent results of program evaluations and quality improvement efforts report in this review.

Despite the frequent identification of needs for staff education and training, only one article focused on evaluating an educational program for screening. Arnold et al. [92] evaluated the effectiveness of a communication skill training program for radiotherapists in using routine screening for distress in Australia. They found that participants in the program increased the “confidence, knowledge and attitudes” of radiotherapists regarding screening and referrals and resulted in an increase in the number of patients screened post-intervention.

#### 3.1.4. Trend: Identifying Screening Benefits

The authors of the included articles in this review clearly supported the need for psychosocial care, emphasized the value of screening for distress as a mechanism to identify individuals at risk for difficulties early, and to take action in referring them for follow-up. However, given the implementation of existing screening initiatives, authors are now describing additional benefits that have been realized to explore the prevalence of distress in various populations and to understand the rate of uptake of referral and intervention [94,95]. Several authors illustrated how access to large datasets, which include screening results, could contribute to understanding factors that contributed to non-screening and the lack of adherence to follow-up [78,87]. Factors such as age, ethnicity, access to technology, and personal willingness of the patient to pursue psychosocial care were reported as key considerations [72,83,87].

Several articles emphasized the need for the on-going evaluation of a screening program and the benefit of having data stored electronically [57,78,79,81,87,96]. Data about program performance (e.g., screening rates, follow-up rates) [48,80,81] as well as levels of distress [93] contribute a basis for practice and for understanding where improvements can be made [46,61,70,89,91]. These data can be utilized to identify gaps in service delivery, for decision making about new initiatives, or as the basis for policy design [21,53,93,96].

### 3.2. Considerations for Oncology Nursing

#### 3.2.1. General Comments

Screening for psychosocial distress continues to be an important aspect of cancer care. It is expected that there will be strategies for screening implemented when a health facility aspires to provide quality whole-person care. However, as facilities move to implement screening for distress programs, challenges exist to greater or lesser degrees depending on the local context and available resources. The evidence is growing about how to overcome these emergent challenges, and there is a clear indication that intentional effort is required to achieve success when implementing efforts to screen for psychosocial distress.

In many cancer care settings, oncology nurses remain key to the implementation of a screening for distress program. Yet, it is noteworthy that no papers were identified in this brief review published in nursing papers about the unfolding lessons regarding successful implementation. While there is evidence that the utilization of symptom distress screening tools can help facilitate nursing interventions and subsequent patient outcomes, this brief review illustrates that there are challenges that arise during the implementation, monitoring, and evaluation of the screening process. These challenges have implications for nurses who are to be involved in screening for distress.

One article evaluated the creation of a nurse-led program for screening for distress [85], and nurses were engaged in reporting barriers and facilitators in several studies [48,75,77,89,90,91]. The information in these articles reflects perspectives similar to those reported in earlier writing about nursing roles and responsibilities for screening for psychosocial distress. This similarity emphasizes that nurses who are starting to implement a screening for distress program can anticipate certain known challenges. However, evidence is mounting about successful strategies to overcome these challenges. Following an implementation science model for introducing innovations in a clinical setting [97] or using a practice change process is strongly recommended [36].

#### 3.2.2. Implications for Practice

The most frequently identified challenge nurses report for implementing a screening for distress program is workload due to high patient–nurse ratios. Nurses report that there is little time for activities related to screening or follow-up of psychosocial issues together with descriptions of shortages in staff numbers or other competing priorities. Unfortunately, nursing shortages exist around the world and are expected to increase [98], driving higher nurse–patient ratios. Yet, at the same time, high-quality patient-centered care demands attention to psychosocial issues. Dealing with this challenge will need leadership and interdisciplinary collaboration [99].

Implementing a program of screening for psychosocial distress in most settings will require practice change not only by nurses but by other members of the interdisciplinary team [91,97]. A critical success factor in achieving the necessary practice change is leadership. The administrative leadership of a healthcare facility must show commitment to the implementation of a screening for distress program, and the nursing leadership must also show commitment and support. Roles and responsibilities will need to be clearly specified for screening and follow-up actions that may require a different workflow and daily practice activities should be implemented. However, the different activities associated with such a practice change can bring about distress on the part of staff members. Without strong leadership for this initiative, clearly stated intentions or expectations of the screening for distress program, and allocation of designated resources and infrastructure, there is potential for failure [36,79].

Drawing from our Canadian experience in various cancer centers in Ontario [100] and across Canada [101], implementing screening for psychosocial distress in a cancer center requires embedding activities related to screening for distress in all aspects of care [21,36]. Clarifying expectations about who will perform the screening and what will happen as a result of the screening procedure is an important first step. Setting out specific expectations for nurses regarding psychosocial assessment, intervention, and referral in their daily practice is necessary. These expectations can be incorporated into the role performance description and included in performance appraisals. In some cancer centers, the model of care is organized so that each nurse discusses the screening results with a patient while, in other centers, there is a triage nurse who engages in an initial conversation about the screening results and directs the patient where to go for assistance.

The second challenge nurses report is feeling they do not have the requisite knowledge and skills to handle screening of psychosocial issues or deal with the resulting concerns identified. Nurses in many countries find they do not have opportunities for education related to specialty oncology knowledge or psychosocial care of cancer patients [102]. Also, the scope of the nursing role varies from region to region with oncology nursing not recognized as a specialty practice or nurses not seen as providing symptom management or emotional care. Educational programs designed to assist nurses with providing screening, basic emotional support, and appropriate referral have been developed and are seen as pre-requisite for implementing a screening for distress program [102,103,104,105].

Because many nurses express discomfort and a lack of preparation to help patients with psychosocial issues, there is a need for educational preparation about psychosocial care prior to any implementation of screening for distress. This educational preparation ought to be heavily weighted toward the assessment of patient needs and provision of basic psychosocial support. Additionally, knowing where referrals can be made is important. Collaborating on the education program with other members of the interdisciplinary team (e.g., social worker, psychologist, spiritual leader, etc.) not only helps the nurses learn but also facilitates team cohesion and clarity of understanding about one another’s roles and expertise.

Additionally, some health facilities have appointed a clinical leader for screening implementation support. Appointing a dedicated person to oversee the process of implementation has been a critical success factor in various cancer centers. This individual needs to have clinical expertise and be available in the clinical arena to help staff members as a screening program is implemented. Ideally, this individual would be a nurse with advanced preparation and be able to work in partnership with other professionals on the healthcare team. The individual would serve as a role model, educator, collaborator, and advocate for psychosocial care. We have observed that, if such a nurse was in the position, this individual would be able to move the project forward through traditional engagement strategies like education sessions, direct mentoring in the clinical setting, role modeling, orientation of new staff, participating in leadership activities related to documentation, oncology nursing role expectations and practice, and performance reviews.

The expectations of other members of the interdisciplinary team for activity related to screening for distress programs should also be articulated. If a nurse is engaged in an initial conversation about screening results with a patient, the nurse needs to understand where a patient can be referred if added assistance is needed. In turn, team members must understand how they are to respond to a referral request, with assessment and/or intervention related to their expertise and the patient’s challenge. The development of clinical pathways for action following screening is important [106].

Incorporating screening for distress variables in the electronic medical record system is also helpful. Although paper-based approaches to screening have been used, a digital system to record clinical observations and intervention recommendations and outcomes, as well as easy data entry and retrieval, is more efficient. A digital system is particularly valid as more services are being delivered virtually versus in a clinical setting. A digital system facilitates documentation as well as data retrieval for audits, quality improvement, or research purposes. Nurses will need to access these data to monitor what is happening with a patient following an intervention and to audit program performance or to identify gaps in service provision.

The implementation of screening for distress needs continued research attention. Understanding how the screening procedures need to be adapted for local contexts remains an important consideration for investigation. Especially with the anticipated shortages of healthcare personnel, it is also important to identify practical approaches for the effective identification of individuals who are at risk of psychosocial distress and would benefit from intervention. Understanding more about approaches for identifying relevant clinical pathways for individuals with different levels of psychosocial distress would be useful. Knowing when a referral is required and which member of the multidisciplinary team ought to be intervening at which levels of distress could also be explored. Other directions for future research could include the identification of essential competencies for screening, a deeper understanding of patient responses to screening, and reasons for declining referral.

## 4. Limitations

The main limitation of this review is that it was aimed as a brief review only, and one database was utilized. No doubt there are other publications about the topic that could have been identified with another design approach (e.g., systematic review, scoping review) or additional search terms, but we believe these articles provide an indication of overall trends surrounding the topic area. Additionally, no quality analysis was completed on the identified articles.

## 5. Conclusions

Screening for distress is recognized as the 6th vital sign in cancer care and is endorsed as part of the provision of quality oncology care. The main trends in research regarding screening for distress include the continued development of screening tools for specific populations and understanding the barriers and challenges to implementation from patient, provider, and cancer system perspectives. The need for the local adaptation of screening procedures is a critical step in achieving success. By continuing to be knowledgeable about the developments surrounding screening for distress, nurses, clinical teams, and cancer, leaders will be able to understand the challenges and barriers regarding implementation and how to overcome them. Nurses have an important role in screening individuals at risk for developing psychosocial problems, providing emotional support and referring patients to members of the multidisciplinary team as necessary to reduce the associated psychosocial morbidity.

## Figures and Tables

**Table 1 healthcare-12-02167-t001:** Summary of selected characteristics of articles reviewed [N = 47].

Domain	Items		Number of Papers	Totals
Region in which author is based	North America	Canada	6	47
		United States	21	
		Mexico	1	
	Europe	Switzerland	2	
		Germany	2	
		Italy	1	
		Spain	1	
	Australasia	China	3	
		Japan	2	
		Vietnam	1	
		Australia	6	
		Sri Lanka	1	
Populations included as focus for articles	Cancer Patients By age groups	Adult	33	40
		Adolescents and young adults	4	
		Pediatric	3	
	Cancer Patents By cancer sites	Mixed cancer sites	18	
		Breast	4	
		Head and neck	3	
		Gynecologic	3	
		Lung/ovarian	2	35
		Pediatric cancers (mixed)	5	
	Cancer Patients Defined by other characteristics	Inpatients	2	
		Outpatients	1	5
		Family caregivers	1	
		Rural dwelling	1	
	Healthcare professionals	Nurses	2	7
		Physicians	1	
		Both nurses and physicians	3	
		Radiation therapist	1	
Type of paper	Commentary	Perspectives on screening for distress approaches	3	3
	Research report	Observational design–quantitative		42
		Retrospective chart audit	9	
		Cross-sectional	4	
		Inception cohort	4	
		Descriptive qualitative	6	
		Mixed methods	1	
		Program evaluation	5	
		Interventional	2	
		Meta-analysis	1	
		Psychometric evaluation/assessment	10	
	Program development	Design/development of program Implementation–describing process only	2	2

**Table 2 healthcare-12-02167-t002:** Barriers and facilitators to implementing screening for distress identified in 47 articles.

	Barriers	Facilitators
Patient-related	Missing capability to express specific needs Changes in emotional wellbeing Embarrassed to reveal distress Feel burdened by questionnaires Do not see value in completing the screening tool Not aware of available resources for help Characteristics may contribute to inability to engage (age, eyesight, language, literacy, cognition, lack of access to electronic equipment/internet, poverty, etc.)	Valuing of holistic care approaches
Clinician-related	Lack of knowledge about psychosocial distress Uncertainty about benefit/effectiveness Workload and time pressures Short stay of patients in facility High turnover/fluctuation of staff Subjectivity of personal motivation Variation in interpersonal expertise Uncertain about ability to respond to patient needs Absence/insufficient continuing education Not seeing incentives to engage with screening Competing pressures or demands	Sufficient knowledge and skills about psychosocial distress Needing little or no change to existing routine processes Adaptability to daily routines
Team-related	Lack of staff Lack of interest by senior staff leaders Priority attention to somatic patient concerns Perceive alignment lies with other healthcare professionals’ assessment/practice Lack of clarity about whose role it is to screen Poor team functioning	Emphasis on teamwork Needing no change in existing referral processes between services
Institutional-related	Lack of ‘buy-in’ by facility Lack of recognition of expertise among staff Lack of required resources (personnel and equipment) Missing objective tool (standardized) Lack of reminders to staff No routine feedback about performance to staff Lack of clarity regarding expectations for screening Lack of clarity regarding routine/process for screening Lack of standardization across facility (screening tools, procedures, data standards, referral criteria) Lack of privacy/space for screening and conversations with patients	Availability of required resources Supportive leadership Feedback on adherence Alignment with existing processes Accountability in IT system Financial benefits are evident
Guidelines and algorithms related	Subjectivity of assessment Missing observability benefits Demands time-consuming patient discussions Lack of clinical pathways for follow-up linked to screening results	Aligned with existing processes Well-tailored to most patients Well defined clinical pathways based on screening results
Policy-related	Stigma of mental health Lack of standardization in procedures (across institutions) including agreement on use of a standardized tool	Interest on part of stakeholders

**Table 3 healthcare-12-02167-t003:** Recommendations made to improve screening for distress in 47 articles.

Trends	Key Recommendations	Articles
Designing and adapting screening tool	Utilize hybrid approach for screening (paper-based, e-screening) (with telemedicine approaches)	Sutton et al., 2022 [72]; DeGuzman et al., 2022 [48]; LaCouteur et al., 2021 [69]
	Use screening instruments that are independent of language	Gunther et al., 2022 [87]
	Utilize a tool with relevant items for the population	Patterson et al., 2021 [54]; Harbeck et al., 2021 [53]; Pepin et al., 2022 [58]; Maldonado et al., 2021 [60]; Brown & Wallace, 2024 [75]; Hirayama et al., 2023 [63]
	Cut-off scores of 4 on the Distress Thermometer can be utilized in various populations	Sun et al., 2021 [47]; Nguyen et al., 2021 [62]; Al-Shaaobi et al., 2021 [45]
Describing programs	Incorporate the screening procedures into the daily routine	Dreismann et al., 2022 [76]; Smith et al., 2022 [77]; Kunz et al., 2021 [51]; Lam et al., 2024 [85]; Rohan et al., 2023 [89]
	Implement an interdisciplinary screening approach	Dreismann et al., 2022 [76]; Rivest et al., 2021 [79]; Vasquez et al., 2022 [74]; Meggiolaro et al., 2021 [80]; Lam et al., 2024 [85]; Brown & Wallace, 2024 [75]
	Collaborate among departments/facilities to facilitate data collection and referrals	Ng et al., 2021 [78]
	Situate screening within a supportive interaction and a stepped or tiered model of care	Seib et al., 2022 [70]; Medina et al., 2022 [81]; Meggiolaro et al., 2021 [80]; Parmet et al., 2023 [73]; Pang et al., 2022 [82]; Zaleta et al., 2023 [68]
	Develop and test relevant interventions that will improve referral uptake for the population	DeGuzman et al., 2022 [48]; Gascon et al., 2022 [50]; Kunz et al., 2021 [51]; Barrera et al., 2021 [55]; Marchak et al., 2021 [57]
Identifying challenges and recommendation for improvement in screening	Work with champions and opinion leaders	Aebi et al., 2023 [91]; Marchak et al., 2021 [57]
	State expectations about role responsibilities clearly	Dreismann et al., 2022 [76]; McCarter et al., 2022 [84]
	Provide clear guidelines for screening procedures and follow-up actions/cut-point scores on screening tool	Dreismann et al., 2022 [76]; Vasquez et al., 2022 [74]; Sun et al., 2021 [47]; Rohan et al., 2023 [89]; Stout et al., 2023 [86]
	Incorporate reminders for staff about screening	Gunther et al., 2022 [87]
	Provide feedback on performance and patient benefits	Aebi et al., 2023 [91]
	Utilize a change management and project management framework to guide implementation of screening programs	Rivest et al., 2021 [79]
	Integration of screening tool, results and documentation into electronic medical patient record	Marchak et al., 2021 [57]; Ng et al., 2021 [78]; Rivest et al., 2021 [79]; Smith et al., 2022 [77]; Rohan et al., 2023 [89]; Medina et al., 2022 [81]; Manikowski et al., 2023 [67]
	Raise awareness through regular training of new employees	Aebi et al., 2023 [91]
	Provide education on psychosocial distress and screening	Dreismann et al., 2022 [76]; Smith et al., 2022 [77]; Vasquez et al., 2022 [74]; Arnold et al., 2021 [92]
	Provide education on communication skills	Dreismann et al., 2022 [76]; Smith et al., 2022 [77]
Identifying screening benefits	Implement a longitudinal approach to screening	Seib et al., 2022 [70]; Smith et al., 2022 [77]; Lacourt et al., 2023 [93]; Brauer et al., 2022 [49]
	Evaluate regularly which patients are not being screened/reasons/contextual factors and if relevant items are missing from the screening tool	Gunther et al., 2022 [87]; Marchak et al., 2021 [57]; Rivest et al., 2021 [79]; Rohan et al., 2023 [89]; Reid et al., 2022 [46]; Maldonado et al., 2021 [60]; Al-Shaaobi et al., 2021 [45]; Harbeck et al., 2021 [53]; Bultz et al., 2021 [71]; Sutton et al., 2022 [72,88]
	Utilize administrative datasets to generate reports for business cases and quality improvement initiatives	Miller et al., 2022 [83]; Gunther et al., 2022 [87]; Sutton et al., 2022 [72,88]; Lacourt et al., 2023 [93]

## Data Availability

Not applicable.

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
