# Peer review of "Screening for Psychosocial Distress: A Brief Review with Implications for Oncology Nursing"

_healthcare, 2024, doi:10.3390/healthcare12212167_

Round 1

Reviewer 1 Report

Comments and Suggestions for Authors

Main comments

Overall I believe this is an interesting and useful brief review of recent literature in an important clinical area. The comments below are mainly suggestions for the authors’ consideration, and minor editing points.

2. Methods: States this is a review of literature from the past two and a half years, but also states it covers March 2021 to July 2024 which is three, or three and a half, years.

2. Methods: I think the search methods are generally sufficient for a “brief review”, but it would be useful to report a little more detail. The search is limited to a search of Medline via PubMed. Also, some of the search phrases are a little restrictive e.g. “screening for distress” and “screening for psychosocial distress” may not pick up other phrases (taken from the references) such as “distress screening”, “screening for emotional distress”, etc. It would be useful to know whether these phrases were searched for in the title and abstract of papers? I am not suggesting re-doing the search in this instance, but for future reference, an information specialist may be able to advice on use of tools such as “adjacent within three words” or wildcards to allow such phrases to pick up a wider range of references. In case any references were missed via the database/keyword search, it would be useful to know whether any additional methods of searching were used, e.g. checking recent relevant reviews, general expert knowledge of current literature, etc?

3. Results: The Distress Thermometer and Problem Checklist is mentioned a lot and it is stated at the start of Section 3 that 23 of 47 articles centred on this. It would be useful to get a flavour of the other tools – how many tools were studied, and were any assessed in several papers, or are most people only using/researching the Distress Thermometer? The start of the section on Special Populations (lines 143-146) suggests that all this section relates to the Distress Thermometer, and the summary paragraph (line 195) also implies this; however, sub-sections on children and caregivers imply that other tools are covered in this section as well, so it may be worth making this slightly clearer.

3.1.2 “Describing programs”: This is a very interesting section since there is little point in screening unless it leads to follow-up interventions. This section is naturally a little limited, since articles about follow-up interventions were not included in this brief review unless part of a program. I wonder if it is worth signposting to the sections on “Recommendations for improvement” and “Implications for oncology nursing” since these sections provide more suggestions regarding the importance of follow-up care?

3.1.3: I would be tempted to split this into two sections for “Challenges” and “Recommendations”, but happy to leave this to the authors’ judgement.

3.1.4 “Identifying screening benefits”: I’m not sure this section relates to the benefits of screening; it appears to relate more to the benefits of improved data collection about screening and referral (whereas benefits of screening may be more along the lines of improved health and quality of life for patients and carers). Perhaps consider whether the theme/title is the best choice?

5. Conclusion: Without making it too long, I wonder if this section could pull out some of the key topics and recommendations from the brief review. Also, perhaps the wording should be tweaked in order to place the responsibility for screening and follow-up on clinical teams and leaders as well as nurses? For example “nurses and xxx are able to understand the challenges and barriers …”; also in the last sentence is it fairer to say that nurses have an important role in screening and referral to interventions, rather than being expected to act to reduce morbidity themselves?

Minor comments on wording and formatting

Throughout: Check for consistency regarding whether “Distress”, “Distress Thermometer”, and “Problem Checklist” need to be capitalised or not.

Abstract: I believe Australasia should be all one word without a hyphen.

1.1 Background: Is the text in brackets before ref 11 required? Should the 1997 report be referenced also?

Table 1: Under health care professionals there are only 7 studies listed. Did the other studies include other professional types, or was this not reported, etc? Also does this refer to the type of professional administering the screening tool?

Table 1: This presents the study characteristics nicely, but is a little confusing regarding which categories are summed together on the right – could this be checked?

Table 1: Should some sections have a grey background? Not sure of the relevance of this formatting.

Line 208: Should this say “that use of virtual tools can present” (omitting “care”)?

Headers in Results section: I find it slightly confusing that the sub-sub headers are bold, since in my mind, bold indicates a higher-level heading, whereas this is a lower level than e.g. 3.1 and 3.1.1. One option may be to split section 3.1.1 into three sections with their own theme, rather than combining them under “Designing and adapting tools”, since each sub-theme is actually a fairly major topic on its own. However, happy to leave this to the authors’ discretion.

Table 2 under “team-related”: should say “whose” not “who’s”.

Table 2: Some of the facilitators may be more intuitive if worded in a more positive way e.g. “little or no change to routine processes” and “No change in referral processes between services” could perhaps be phrased as “Consistent routine processes” and “Consistent referral processes between services”?

Table 3: Is it possible for the header to appear on the same page as the table?

Line 349: There is a “(ref)” in brackets that needs omitting or adding.

Line 376: Should say “where” not “were”.

Line 385: Spelling of “pre-requisite”

Acknowledgements: This section needs to either be omitted or replaced with the relevant acknowledgements.

Reviewer 2 Report

Comments and Suggestions for Authors

Well-written and relevant manuscript. The content of the manuscript algins with the stated purpose. Minor suggestions are listed below. 

Line 74: correct spelling of center

Lines 101-103: It seems to be the exclusion criteria. If so, it does not need to be in parenthesis. 

Line 145: This is the first place the problem checklist is mentioned. It would be beneficial to describe what it is. 

Line 178-179: problem list or problem checklist?

Reviewer 3 Report

Comments and Suggestions for Authors

The authors address a very important topic in oncology. The focus here is on the screening of distress. The background is explained in a very detailed introduction. I would recommend shortening and concentrating on the essentials, for example lines 61 to 71, 77 to 84.

In the methods section you write that the review covers the past 2 and a half years, but later you report a period from March 2021 to July 2024. How does that fit together?

You have written: Articles with a focus only on reporting levels of psychosocial distress in a patient sample, symptoms of distress, or interventions to manage distress were not included. However before: All article types were considered (e.g., reviews, perspectives papers, descriptive/intervention studies)….. Do the reviews fulfil your inclusion criteria?

 I don't quite understand whether intervention studies were allowed to be included or not? Can you please clarify that?

 Please place a full stop at the end of the sentence in line 108.

 Are there any details about who carries out the screening and when?

 This is just a general note: the many subsections make the paper somewhat confusing. The implications for oncology nursing also fit well in a discussion section.

 I would like the implications to be formulated a little more clearly.

In general, this is a very interesting work that would benefit from a revision of the points mentioned.

Reviewer 4 Report

Comments and Suggestions for Authors

 Dear authors,

the paper is relevant for the nursing profession and the care of oncology patients. The results are presented in detail, the implications for the nursing profession are elaborated.

In the description of the methods, it is necessary to state in detail how the papers included in the review were analyzed - how many authors and how they analyzed the papers, how many papers were found in total, whether all papers were included in the analysis, how the quality of the papers were assessed

It is necessary to check the number of papers included in the title of table 1

It is necessary to provide directions for future research - what topics need to be researched

Expand the limitations of the review – only one database was used, was the quality of the works analyzed,

Comments on the Quality of English Language

Minor editing of English language required.

Round 2

Reviewer 3 Report

Comments and Suggestions for Authors

Many thanks for the careful revision.